# A simple pyrocosm for studying soil microbial response to fire reveals a rapid, massive response by *Pyronema* species

**Thomas D. Bruns** [1]*, **Judy A. Chung**[2], **Akiko A. Carver**[1], **Sydney I. Glassman**[2]

**1** Department of Plant and Microbial Biology, University of California, Berkeley, California, United States of America, **2** Department of Microbiology and Plant Pathology, University of California—Riverside, Riverside, California, United States of America

* pogon@berkeley.edu

## Abstract

We have designed a pyrocosm to enable fine-scale dissection of post-fire soil microbial communities. Using it we show that the peak soil temperature achieved at a given depth occurs hours after the fire is out, lingers near this peak for a significant time, and is accurately predicted by soil depth and the mass of charcoal burned. Flash fuels that produce no large coals were found to have a negligible soil heating effect. Coupling this system with Illumina MiSeq sequencing of the control and post-fire soil we show that we can stimulate a rapid, massive response by *Pyronema*, a well-known genus of pyrophilous fungus, within two weeks of a test fire. This specific stimulation occurs in a background of many other fungal taxa that do not change noticeably with the fire, although there is an overall reduction in richness and evenness. We introduce a thermo-chemical gradient model to summarize the way that heat, soil depth and altered soil chemistry interact to create a predictable, depth-structured habitat for microbes in post-fire soils. Coupling this model with the temperature relationships found in the pyrocosms, we predict that the width of a survivable "goldilocks zone", which achieves temperatures that select for postfire-adapted microbes, will stay relatively constant across a range of fuel loads. In addition we predict that a larger necromass zone, containing labile carbon and nutrients from recently heat-killed organisms, will increase in size rapidly with addition of fuel and then remain nearly constant in size over a broad range of fuel loads. The simplicity of this experimental system, coupled with the availability of a set of sequenced, assembled and annotated genomes of pyrophilous fungi, offers a powerful tool for dissecting the ecology of post-fire microbial communities.

## Introduction

Fire is a natural part of many ecosystems, and organisms in systems with predictable fire regimes are often well adapted to survive or recolonize rapidly after fire. Plant adaptions are particularly well known with the ability to survive fire through thickened bark, serotinous pine cones, vegetative resprouting, or other traits [1]. Soil microbes also must survive or recolonize

---

**Data Availability Statement:** The underlying data for the work consisting of: temperature and fuel data for all pyrocosms and the complete OTU table representing fungal community response. These

---

are deposited in Dryad: doi:10.5061/dryad.
45gd695; Representative sequences for all the
OTUs are deposited in NCBI MN724033-
MN724919, and the raw sequence read data are
deposited in the short read archive: PRJNA559408.
R-scripts for all sequence analyses are deposited in
Github: https://github.com/sydneyg/Pyrocosms.

**Funding:** The work was funded by the Department
of Energy grants DE-SC0016365 and DE-
SC0020351 to TDB. The funders had no role in
study design, data collection and analysis, decision
to publish, or preparation of the manuscript.

**Competing interests:** The authors have declared
that no competing interests exist

after fires, but much less is known about how they achieve this or what their roles are in the post-fire environment. Pressler et al. [2] conducted a meta-analysis on belowground effects of fire and reported that negative effects are commonly found on microbes including reductions in biomass, abundance, richness and evenness across taxonomic groups, and these effects were coupled with decade-long recovery times. Fungi in particular showed large reductions and slow recovery [2]. Dove and Hart [3] found similar effects in their meta-analysis of fungal communities, and both studies showed that the largest effects were associated with forest biomes. Post-fire changes in fungal communities were thought to be caused by both the direct killing effects of fire on the organisms, and by indirect effects on habitat, such as the loss of the organic layers and the reduction in living root biomass of mycorrhizal hosts.

Almost all studies considered in these meta-analyses focused on the reduction in post-fire microbial communities, but there is also evidence of microbes that respond positively to fire. In particular there is a set of "**pyrophilous**" fungi that fruit only in burned habitats and are abundant in the first weeks or months following fire. These predominantly saprophytic fungi have been known for over a hundred years [4], and the pattern of their fruiting in post-fire settings suggests a rapid successional sequence [5, 6]. Nevertheless, the rapid, post-fire, fungal succession has not been confirmed at the mycelial level with modern, molecular methods, nor is it known what these fungi degrade and live on in the chemically altered post-fire environment. How these fungi rapidly colonize post-fire soil is also unknown. Some fungi have heat-stimulated spores [7], or spores that can also be stimulated by post-fire chemicals [8]; these propagules may reside in the soil for decades between fire events [9]. Other post-fire fungi have been shown to colonize plants endophytically [10], but how this translates into colonizing the burnt soil habit is unresolved.

We propose that the post-fire microbial habitat is structured by a thermo-chemical gradient in which direct heating effects of fire and the production of temperature-specific soil chemicals change predictably with soil depth and fire intensity. The two main components of this conceptual model are: 1) a steep temperature gradient that causes differential mortality with soil depth; and 2) a gradient of chemically-altered substrates produced by this temperature gradient that structures resources available to recolonizing microbes. Fire heating of soils is reasonably well understood from a variety of models [11–13] and is the basis for component 1. These studies have shown that the heat capacity and water content of soil produce depth-stratified temperatures where high temperatures at the surface drop rapidly with depth. We assume that the temperatures achieved in soil will cause a gradient of death in which high temperatures near the surface kill all organisms down to some threshold depth, and below this depth there will be differential survival. The lethal temperature varies with the organism [14], and those organisms tolerant of higher temperatures will find themselves in a zone of reduced competition. We call this the "**Goldilocks zone**", a depth layer where the not-too-hot, not-too-cool temperatures allow fire-specialized microbes such as the pyrophilous fungi to survive while killing most competitors. At least some pyrophilous fungi appear to have dormant spores or sclerotia in the soil, and these germinate following heating or chemical changes associated with fire [6, 15]. We would expect a rapid stimulation of such propagules within the Goldilocks zone.

The effects of fire on soil chemistry are known to be correlated with the temperature/depth gradient in general ways, and this knowledge is the basis of component 2 of our model. For example, extreme surface temperatures can completely combust much of the ligno-cellulosic biomass into $CO_2$, while producing a cation-rich, high pH ash. Substantial amounts of biomass are also transformed into partially pyrolyzed (i.e., heat-modified), highly aromatic carbon sources [16]. Temperatures from 220–480 ˚C are known to convert biomass into a mixture of partially pyrolyzed organic compounds, while volatilized waxes and lipids typically condense

at temperatures around 200˚C in the soil below[16]. This process results in a hydrophobic zone that commonly occur in forest fires and can lead to heavy erosion by channelizing runoff [17–19]. Thus, the upper layers of soil will have highly modified, fire-specific carbon sources that may select for specific sets of organisms able to metabolism them. At lower depths, where peak temperature is below 200 ˚C, little pyrolysis occurs, but the high temperatures still kill most life. This creates a **necromass zone,** where dead organisms are likely to provide easily-mineralizable forms of carbon and nitrogen, creating a nutrient subsidy for any microbes that can rapidly recolonize. This zone is immediately above the Goldilocks zone and intergrades into it with the differential survival of heat tolerate microbes.

Experimental evidence of the effects of fire on microbial communities has been focused on sampling studies of wildfires and prescribed burns (see studies included in [2, 3]). Although generalities have been learned from these approaches, they have not been useful for connecting the detailed understanding of soil heating and chemistry to the structure of post-fire microbial communities. Post-fire successional studies of wildfires have generally been space-for-time comparisons with limited replication, and sampling was usually not conducted until at least a year or more post-fire [2,3]; thus the early responses would be missed. Prescribed burns have more potential for pre and post-fire sampling, replication, and fuel manipulations. Petersen's [6] early work on successional patterns of pyrophilous fungi involved small experimental fires but were conducted prior to molecular identification methods and relied on fruiting. However, recent work by Reazin et al. [20] used replicated fuel manipulations, pre- and post-fire sampling, and high-throughput sequence analysis to show a strong response by pyrophilous fungi that varied with fire intensity. Nevertheless, all prescribed fires are necessarily limited to times when air temperatures, moisture, and other weather parameters make control manageable, and prescribed fires can only be manipulated in a relatively course way to achieve differences in soil temperatures. They also involve significant site preparation and costs that typically limit studies to small numbers of sites.

Our goal was to develop a more easily manipulated system to allow us to control soil heating and to dissect the fire effects on soil microbes at a finer scale. To that end we have developed a "**pyrocosm**" system where key fire effects on soil microbial communities can be controlled, monitored, and replicated in a few liters of soil. Here we show that we can use this system to control and reproduce soil heating effects by manipulating coarse fuels, and we use the results of these experiments to further develop our model of the Goldilocks and necromass zones. As a proof of concept we use forest soils to show that known pyrophilous fungi can be rapidly simulated to grow in these systems, and we discuss ways in which the growing genomic knowledge of these fungi can be used to further dissect post-fire, fungal ecology.

## Materials and methods

### Pyrocosm design

Our pyrocosms consisted of 10-quart (9.46 l) galvanized steel buckets, filled to a depth of 16cm with test soil or sand, and wired with K-type thermocouples at selected soil depths through small drill holes in the bucket sides. Total volume of soil in the buckets was 7 liters, which left a unfilled margin of approximately 8.5 cm at the top. Ten to 13 drill holes (0.6 cm diameter) were added to this unfilled margin to increase aeration for the fire. Assembled units were buried so that soil levels inside and outside the bucket were even. A small fire with weighed fuels consisting of pine needle litter, paper and charcoal briquettes was then burned on the top of it (Fig 1) and the test soil was watered the next day with to initiate microbial activity after it had completely cooled. Detailed notes on assembling these pyrocosms are given in the (S1 File)

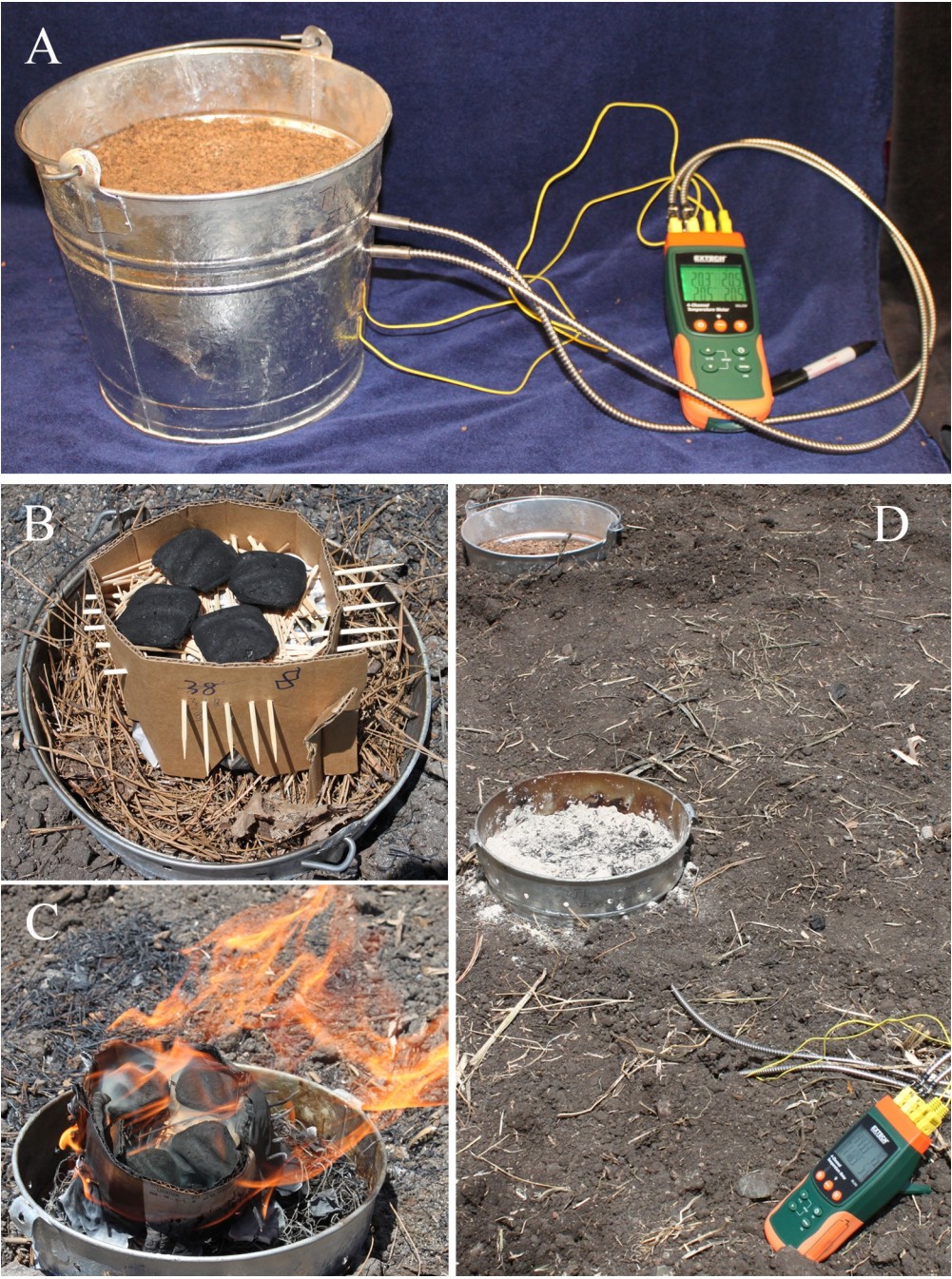

**Fig 1. Pyrocosm construction.** A) Pyrocosm in the lab filled with forest soil and wired with thermocouples; air holes around rim had not yet been drilled but are seen in remaining images; B) weighted pine needles, cardboard, tooth picks and newspaper constitute "flash fuels" used to ignite charcoals; C) early ignition phase; D) burned pyrocosm and unburned control in background.

Two pyrocosms were used to study post-fire fungal response and were filled with forest soil. Unburned test soil for these was collected from a *Pinus ponderosa* forest in Stanislaus National forest, CA, USA, 37.8140–120.0689, just outside the perimeter of the 2013 Rim Fire. The top 5 cm and the bottom 5–15 cm of soil were collected and kept as separate samples. Naturally

burned test soil was collected from a nearby, fire-killed, *Pinus ponderosa* forest within the perimeter of the Rim fire in Stanislaus National forest, 37.8442–119.9402, located adjacent to previously studied plots [21], and was not depth stratified because it was derived from a disturbed pile. All soils were collected in Spring 2015, under Special Use Permit #GRO1087 from the USDA Forest Service, Stanislaus National Forest to TDB. Litter and F layers were collected from the same unburned site. The soils were sieved through a 2mm soil screen to homogenize them and remove rocks, roots and large aggregates, and then were air-dried in large, closed paper bags for 3 weeks in a fume hood to a moisture content of between 3.8% (bottom 5–15 cm) and 4.8% (top 5 cm), and then stored in sealed plastic bags at 5˚ C until needed. When assembling soil pyrocosms, bottom soil was used for the lower 5–16 cm and top soil was used for the upper 5 cm. In the second soil pyrocosm, burned forest soil was mixed with the bottom unburned soil in a 1:9 ratio with intent of adding inoculum of post-fire fungi to the experiment. Soil was allowed to settle evenly, but was not compressed. Bulk density was not measured, but was likely greater than in the field because sieving destroys structure that would otherwise produce larger air pockets.

Course sand (~2mm grain) was used instead of soil in the series of 14 burning experiments to test the relationship between fuel and soil temperatures. Sand was selected because it was not altered by the fire, and therefore pyrocosm experiments could be repeated without reassembling the units between fires. Instead, the ash and charcoal formed were simply removed by hand and with the assistance of a small fan, and then a new fuel load and heating experiment were tested. Thus all 14 experiments were run with three sand pyrocosms. Controls for these temperature experiments consisted of leaving one of these units unburned. This gave us a reading of diurnal soil temperature changes unrelated to the fires.

Thermocouples were K-type, with an accuracy of 1˚C. In the two soil pyrocosms, four thermocouples were installed through drilled holes in the side of the buckets at 0.5, 3.5, 6.5, and 9.5 cm from the surface (Fig 1A). These were inserted as the soil was added such that the tip of each thermocouple was located near the center of the bucket at the prescribed depths (see S1 File for detailed notes). The top two thermocouples were insulated K-type thermocouples that could withstand temperatures up to 1038˚C (custom made by OMEGA.com, see Fig 1A). The lower two were plastic insulated wire thermocouples with temperature maximums of 200˚C (Fig 1A). In later experiments, the number and arrangement of thermocouples per pyrocosm were modified as needed for specific uses. For experiments designed to determine the effect of fuel load on peak temperatures only two thermocouples were used and placed at 10.5 and 15.8 cm from the surface. These were the plastic-insulated types since temperatures were more moderate at these depths and the thinner wires were more accurately placed. These depths were selected to insure that the wire thermocouples did not experience temperatures above 200˚C across the range of fuel loads tested. Temperatures were recorded using a data logger (Extech Instruments SDL200) at 10 sec or 30 sec intervals. The 30 sec intervals turned out to be more than sufficient because temperatures change fairly gradually at the depths measured.

## Fuels, ignition and fire

Fuel in all experiments was weighed prior to ignition. Dried, weighed, forest litter and F-layer collected from the same unburned site as the soil were placed at the soil surface and gently tamped down to achieve an approximate depth of approximately 4 cm, which was similar to the depth of the organic layers at the site where the soil and litter were collected. Litter and F-layer samples was re-dried the night before in a drying oven at moderate heat (~110 C) to ensure that it burned readily and was consumed completely during the experiments. Charcoal was used as the course-fuel for these experiments, because it burns uniformly and completely

once ignited. Differences in heating were achieved by simply varying the number of charcoal briquettes, which were weighed prior to burning, and were placed on top of the litter to ignite them (Fig 1). Our source of charcoal was a local grocery store.

The fire for the two soil pyrocosms contained approximately 300 g of litter and 1000 g of charcoal. The charcoal was piled in the middle of bucket, and was spread evenly after it was well lit. The temperatures at the four depths were monitored and the charcoal was then carefully removed with a small trowel when soil reached a target temperature. The removed charcoal was extinguished rapidly by dropping it in water, and was later dried and weighed and used to determine the amount that had been consumed. The temperature was monitored for six more hours and pyrocosms 1 and 2 ultimately reached temperatures of 146˚C and 171˚C, respectively, at a depth of 6.5 cm. The temperatures from these soil experiments were analyzed separately from the sand pyrocosms, since the heat capacity of soil and sand was assumed to differ.

In all sand experiments the amount of charcoal was reduced to 1 to 15 briquettes (~25 to 380 g) and was ignited by perching it on top of fine wood kindling (tooth picks), held in a 28–38 g cardboard ring that was filled with a single sheet of newspaper (Fig 1B). After the cardboard ring burned, and the ignited charcoal was moved toward the center, spread evenly, and allowed to burn completely. This complete combustion of a reduced amount of charcoal allowed us to determine the effect of charcoal quantity on peak soil temperatures more easily.

## Fungal incubation, sampling, and controls

The two pyrocosms filled with forest soil were used to study fungal response to fire. The day following the experimental fires, after the soil temperatures had returned to normal, water was added to the pyrocosms in the form of weighed, crushed ice. This method for adding the water was meant to mimic snow, which is a typical way late fall precipitation occurs at mid elevation in the Sierra Nevada Mountains where the soil was collected. The slow melting also allowed the water to gently and uniformly percolate into the soil, even when the soil had become slightly hydrophobic from the fire. Pyrocosm 1 received 1000 g of ice on day 1, and 500 g on day 2, Pyrocosm 2 received 1500 g of ice on day one and 1,500g on day 3. This corresponds to 33 and 66 mm of rain for the two pyrocosms, respectively. The watering differences were meant to provide a broader range of post-fire conditions in these pilot studies with the intent that at least one of them might stimulate pyrophilous fungi. A control pyrocosm, filled with the same forest soil, setup in the same way and buried a meter from the test pyrocosm 1, was left unburned, but watered and incubated in the same way as the first pyrocosm (Fig 1D). After watering, pyrocosms were covered with foil to prevent additional precipitation and to limit aerial dispersal, and they were incubated *in situ*.

Soil within the pyrocosms was sampled from the soil surface to the bucket bottom with a single 2 cm diameter soil corer at each time point. Each soil core was vertically divided into four approximately even depth zones, and DNA was extracted from each zone sample via MoBio DNAeasy Power Soil kit (Qiagen, Carlsbad, CA, USA). Both experimental pyrocosms and the control were sampled this way at weeks 1 and 2 post-fire. In addition pyrocosm two was sampled at week 4.

## Construction of Illumina libraries, and processing of Illumina data

We amplified ITS1 spacer, which is part of the Internal Transcribed Spacer region, the universal DNA barcode for fungi [22], with the ITS1F-ITS2 primer pair using Illumina sequencing primers designed by Smith and Peay [23], and we prepared libraries for Illumina MiSeq PE 2 x 250 sequencing as previously described [21]. These primers were used to enable direct

comparison with earlier sequence studies that included these sites where the soil used was collected. Sequencing was performed at the Genome Center at the University of California, Davis, CA, USA. Bioinformatics was performed with UPARSE [24] usearch v7 and QIIME 1.8 [25] with the same methods as previously published [21]. All analyses are based on 97% sequence similarity for operational taxonomic units (**OTUs**) and taxonomy was assigned with the UNITE fungi database [26] accessed on 30 Dec 2014. Representative sequences for all OTU that represented 1% or more of the read abundance in either of the pyrocosms or the control were BLASTed [27] individually against the NCBI database, and the results were examined to improve the automated sequence-based identifications. Samples for a related study were run at the same time and OTUs were processed for both simultaneously to enable later comparison. As a result OTU identifying numbers reported here are greater than the 887 total found in this study. Sequences are available at the NCBI Sequence Read Archive (PRJNA559408), and representative sequences from each of the OTUs are deposited in NCBI MN724033-MN724919.

A mock community composed of equal quantities of DNA extractions of *Coprinellus sp. Pholiota sp. Cyathus stercoreus*, *Penicillium citreonigrum*, *Aureobasidium pullulans*, *Rhodotorula mucilaginosa*, *Cladosporium sp*., *Suillus sp*., *Peyronellaea glomerata*, and *Peyronellaea glomerata* was included to control for OTU processing, and a no DNA control was included to assay for spurious contamination and index jumping [28].

## Statistical analyses

Plotting of temperature changes over time was initially done with Microsoft Excel, and then repeated in R along with all modeling and correlation analyses [29]. The response of individual OTUs was investigated by examining their changes in % sequence relative to the unburned control.

To determine overall abundance ranks of OTUs, the sequences across all depths in weeks one and two were separately summed for each OTU for each of the two pyrocosms and the control, and divided by total sequences of all OTUs from the same experimental unit and time sample (i.e., pyrocosm 1, 2 or control and weeks 1 or 2). The spreadsheet was then sorted by sequence abundance, first for the control and then for each of the two experimental pyrocosms. Ranks were assigned in descending order within each experimental unit with the most abundant OTU as 1.

OTU increases or decreases relative to fire were investigated by subtracting percent sequence abundance for each OTU from each pyrocosm from the percent sequence abundance in the control. The OTUs were then sorted by this metric to identify those that showed the largest changes.

## Results

### Soil heating characteristics in the pyrocosms

The heating experiments showed the following about the peak temperature that is reached at a given depth: 1) The reproducibility of the temperature profiles is excellent when fuel levels are constant (Fig 2A). 2) Peak temperatures at depth are reached hours after the fire has gone out and they linger near the peak for 40 minutes or more (Fig 2A). 3) Peak soil temperatures decrease by the natural log of the depth (Fig 2B) and can be predicted by the mass of course fuel (Fig 2C). 4). There are edge effects that result in progressively cooler temperatures away from the center (Fig 2D).

A regression between peak temperatures and ln of depth was linear with a $R^2$ of 0.9977 (Fig 2B). The predictability of peak temperatures and total fuel is shown as a polynomial model

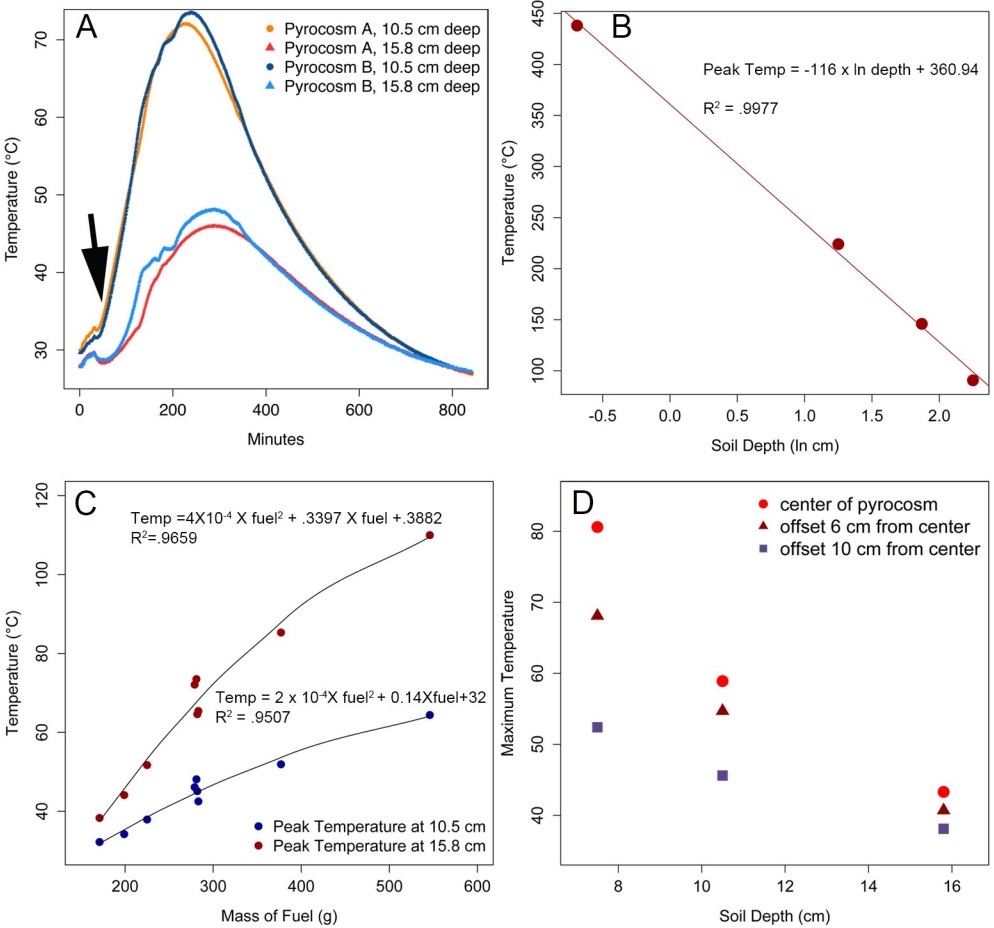

**Fig 2. Thermal characteristics of pyrocosms.** A) temperature profiles from two depths of two fuel-replicated pyrocosms. Half-minute intervals are plotted over 800 minutes. Note that the temperature profiles are quite repeatable, and that temperature continues to rise after the fire is out (arrow); B) peak temperatures are predicted by the ln (or log10, not shown) of the depth, and C) by the mass of charcoal; D) peak temperature at three depths (7.5, 10.5, 15.8 cm) and three locations offset from center within a depth. Note that the center is hottest and the edge of the unit is cooler. Results shown in A,C, & D are from sand pyrocosms; B is from forest soil pyrocosm 1.

with $R^2$ values of 0.9682 (10.5 cm depth), and 0.9507 (15.8 cm depth), but a simpler logarithmic model works nearly as well with $R^2$ values of 0.9659 and 0.9478, respectively.

Flash fuels, such as pine needles, tooth picks, cardboard, and newspaper, that produced few or no coals, had almost undetectable heating effects on the soil at depth. Burning of 172 g of flash fuels caused a 2° C rise in temperature at 10.5 cm below the surface, relative to the unburned control. This contrasts with approximately an 8° C rise at the same depth when a single 25.5 g charcoal was added. However, the flash fuels did cause a more rapid rise in temperature than that caused by solar heating in the control (S1 Fig), and using regression from the measured temperatures points (i.e., Fig 2B), the temperature 1 mm from the surface was predicted to have peaked at 108° C, and soil temperature was predicted to be heated to 70° C at a depth of 1.26 cm.

There is a significant edge effect on soil heating in the pyrocosms, and it is most pronounced at the shallower depths (Fig 2D). For example, at 7.5 cm the peak temperature varied from 91 to 61 ˚C, depending on whether the temperature was measured at the center or near the edge of the pyrocosm. Near the bottom of the unit at 15.8 cm, the temperature from center

to edge only varied 5 ° from 48 to 43 ° C, which is a lesser absolute and relative difference than the variation recorded at 7.5 cm. Spreading out the coals more did not greatly reduce this edge effect (data not shown). The edge effects do not affect the results presented above with respect the relationships between depth and mass of charcoal on peak temperature, or the temperature-time profiles except to limit their accuracy to the center of the units. However, these edge effects do mean that peak temperature isoclines would be curved upward on the edges.

## Fungal response to the pyrocosms

Table 1 summarizes the read depth, number of OTUs, and ranked abundance of the all OTUs in experimental and control samples that had sequence abundance of 1% or greater in control or treatment pyrocosms. Four of the 20 experimental samples yielded 58 or fewer sequences and were dropped. All other experimental samples yielded more than 3950 sequences and were retained. Because the dropped samples made the depth sampling difficult to compare across time, data from depth samples for single time points within an experimental unit were combined to give more robust measures of fungal communities within a time point for Table 1.

The fungal communities in the forest soil pyrocosms showed a fire response that is consistent with stimulation of *Pyronema* spp., a reduction in richness and evenness of the community, but little apparent change to underlying composition (Table 2). Non-metric multidimensional scaling analysis shows that the two pyrocosm communities are about as distinct from each other as they are from the unburned control and also shows much variation occurs between individual depth samples and time points (S2 Fig). However by sorting the OTU table by taxa that increase in the pyrocosms relative to the fire, three OTUs were found that exhibited dramatic increases in both pyrocosms relative to the control. All three were identified as *Pyronema* species (Table 2), a genus that is well known for its rapid fruiting following fire [4, 6].

The three *Pyronema* OTUs were highly abundant in the burned soils, but not the controls (Table 2). *Pyronema domesticum* (OTU7), was the most abundant taxon in Pyrocosm 1, accounting for 57.8% of the sequence in weeks 1 and 2 combined, and was the second most abundant sequence in Pyrocosm 2, accounting for 10.85% of the sequence. *Pyronema omphalodes* (OTU 5234) was the second most dominant taxon in Pyrocosm 1, accounting 6.81% of the total sequence, and it was the most dominant taxon in Pyrocosm 2 in which it accounted for 37.02% of the aggregate sequence in the first two weeks. A third *Pyronema* (OTU 4131) was ranked 10th and 4th in abundance in Pyrocosms 1 and 2, respectively. All three *Pyronema* OTUs showed rapid rises, and declines in relative sequence abundance within the short time span of the experiment (Fig 3A). In addition, *Pyronema domesticum* (OTU7) fruited on the surface of the soil after 17 days in Pyrocosm 2, even though it was less abundant than *Pyronema omphalodes* (OTU 5234) in that experiment (Fig 3). All three *Pyronena* OTUs were detected in the control, but their read abundances were orders of magnitude lower and they did not increase with time as was the case in the burned pyrocosms (Fig 3A). *Pyronema*

**Table 1. Summary of samples, read depth, and OTUs.**

| | Unburned Control | Pyrocosm 1 Weeks 1 & 2 | Pyrocosm 2 Weeks 1 & 2 | Pyrocosm 2 week 4 | Mock control | No DNA control |
|---|---|---|---|---|---|---|
| # samples retained (and lost) | 8 | 7(1) | 6 (2) | 3(1) | 2 | 4 |
| Ave sequence depth/sample | 54427 | 67308 | 25247 | 8495 | 21087 | 408 |
| Range of sequence depth | 9639–71080 | 37014–109855 | 6607–68272 | 3950–15637 | 16530–25643 | 90–1044 |
| # OTUs $\geq$ 0.01% sequence | 406 | 158 | 155 | 110 | 24 | 20 |
| # OTUs with read depth > 5 seq | 605 | 296 | 191 | 164 | 20 | 16 |
| % reads of most abundant OTU | 6.55 | 57.78 | 37.02 | 45.01 | 64.34 | 31.72 |

**Table 2. Most abundant OTUs summed across first two weeks.**

| OTU | Assigned Taxon[1] | Control | | Pyrocosm 1 | | Pyrocosm 2 | | Inferred Ecology[4] |
|---|---|---|---|---|---|---|---|---|
| | | Rank[2] | %seq[3] | Rank | %seq | Rank | % seq | |
| 7 | *Pyronema domesticum* | 30 | 0.85 | 1 | 57.78 | 2 | 10.85 | pyrophilous fungi |
| 5234 | *Pyronema omphalodes* | 165 | 0.05 | 2 | 6.81 | 1 | 37.02 | pyrophilous fungi |
| 4131 | *Pyronema aff. omphalodes* | 315 | 0.01 | 10 | 1.18 | 4 | 7.04 | pyrophilous fungi |
| 57 | *Sarocladium kiliense* | –– | 0.00 | –– | 0.00 | 6 | 5.66 | Soil saprobe, opportunistic pathogen |
| 149 | *Cortinarius sp.* | 37 | 0.24 | 26 | 0.25 | 11 | 1.09 | ectomycorrhizal symbiont |
| 37 | Unknown Geminibasidiales | 1 | 6.60 | 5 | 3.16 | 3 | 10.75 | xerotolerant yeast (Basidiomycota) |
| 61 | Unknown Helotiatles | 2 | 5.22 | 13 | 0.72 | 30 | 0.19 | saprobe/ plant pathogen/ endophyte |
| 42 | *Solicoccozyma sp.* | 3 | 4.25 | 3 | 4.59 | 12 | 1.02 | yeast (Basidiomycota) |
| 5444 | *Mortierella sp.* | 4 | 4.08 | 9 | 1.46 | 28 | .27 | soil saprobe/root endophyte |
| 73 | *Russula acrifolia* | 5 | 3.21 | 4 | 4.58 | 5 | 6.19 | ectomycorrhizal symbiont |
| 89 | Unknown Sporobolales | 6 | 3.03 | 14 | 0.66 | 10 | 1.42 | yeast (Basidiomycota) |
| 5 | *Mortierella sp.* | 7 | 3.03 | 7 | 2.68 | 26 | 0.27 | soil saprobe/root endophyte |
| 161 | Unknown Basidiomycota | 8 | 2.86 | 81 | 0.03 | 132 | 0.01 | Unknown |
| 112 | *Hydropus sp.* | 9 | 2.30 | NR | 0.00 | NR | 0.00 | Saprobe (Basidiomycota) |
| 106 | *Wilcoxina aff. rehmii* | 10 | 2.08 | 19 | 0.33 | 34 | 0.14 | Ectomycorrhizal symbiont |
| 75 | *Pseudeurotium* sp. | 11 | 2.06 | 12 | 0.98 | 8 | 1.87 | soil saprobe/root endophyte |
| 102 | *Sebacina aff. dimitica* | 12 | 2.04 | 32 | 0.15 | 45 | 0.07 | possible ectomycorrhizal symbiont |
| 100 | *Solicoccozyma terricola* | 13 | 2.00 | 11 | 1.05 | 20 | 0.58 | yeast (Basidiomycota) |
| 114 | Unknown fungus | 14 | 1.91 | NR | 0.00 | NR | 0.00 | unknown |
| 115 | *Hyaloscypha sp.* | 15 | 1.87 | 39 | 0.09 | 131 | 0.01 | saprobe/ endophyte |
| 129 | *Ciliolarina sp.* | 16 | 1.86 | 30 | 0.16 | 130 | 0.01 | saprobe/ plant pathogen/ endophyte |
| 132 | Unknown Helotiatles | 17 | 1.83 | –– | 0.00 | NR | 0.00 | saprobe/ plant pathogen/ endophyte |
| 195 | Phialocephala sp. | 18 | 1.77 | 22 | 0.32 | 81 | 0.02 | dark septate root endophyte |
| 190 | Unknown Basidiomycota | 19 | 1.73 | NR | 0.00 | –– | 0.00 | unknown |
| 145 | Unknown Ascomycota | 20 | 1.68 | 67 | 0.04 | NR | 0.00 | unknown |
| 142 | Unknown Pleosporales | 21 | 1.52 | NR | 0.00 | NR | 0.00 | saprobe/ plant pathogen/ endophyte |
| 4306 | Unknown Geminibasidiales | 22 | 1.21 | 15 | 0.63 | 7 | 2.49 | xerotolerant yeast (Basidiomycota) |
| 52 | *Solicoccozyma terreus* | 23 | 1.15 | 6 | 2.92 | 9 | 1.85 | yeast (Basidiomycota) |
| 65 | Unknown Basidiomycota | 24 | 1.13 | 8 | 1.81 | 124 | 0.01 | unknown |
| 2145 | Helotiales sp | 25 | 1.02 | 33 | 0.14 | 67 | 0.03 | saprobe/ plant pathogen/ endophyte |

All taxa represented by at least 1% of the total sequence reads in the control or either experimental pyrocosm are listed.

[1] Assigned taxa are based on automated assignments followed by individual Blast matches and evaluations; detailed information on the behavior of individual taxa in each sample is give in the deposited OTU table (doi:10.5061/dryad.45gd695).

[2] ranked order of OTUs based on sequence abundance, where 1 is the OTU with the most sequence reads in that treatment.

[3] % of sequence reads assigned to the taxon in the control and each experimental pyrocosm within the first two weeks combined. First five taxa are those which are abundant in the pyrocosms, but not in the control, remaining 25 are ordered by their abundance in the control.

[4] inferred ecology is based on knowledge of the taxon at the generic level. Data for Pyrocosm 2 is restricted to the first two weeks to allow comparison with the control and pyrocosm 1. NR = present but not ranked, below 0.01%.–– = not found.

*domesticum* (OTU7) was the most abundant of the three in the unburned control, where it accounted for 1.69% in the first week but declined to 0.01% by week two. As discussed below it also appeared in the mock community control presumably via index switching [30] and accounted for 0.22% of the sequences (92 reads).

Only 79 OTUs had sequence abundance higher than 0.1% in at least one of the two experimental pyrocosms. Of these, 38 OTUs showed nominal increases in sequence abundance

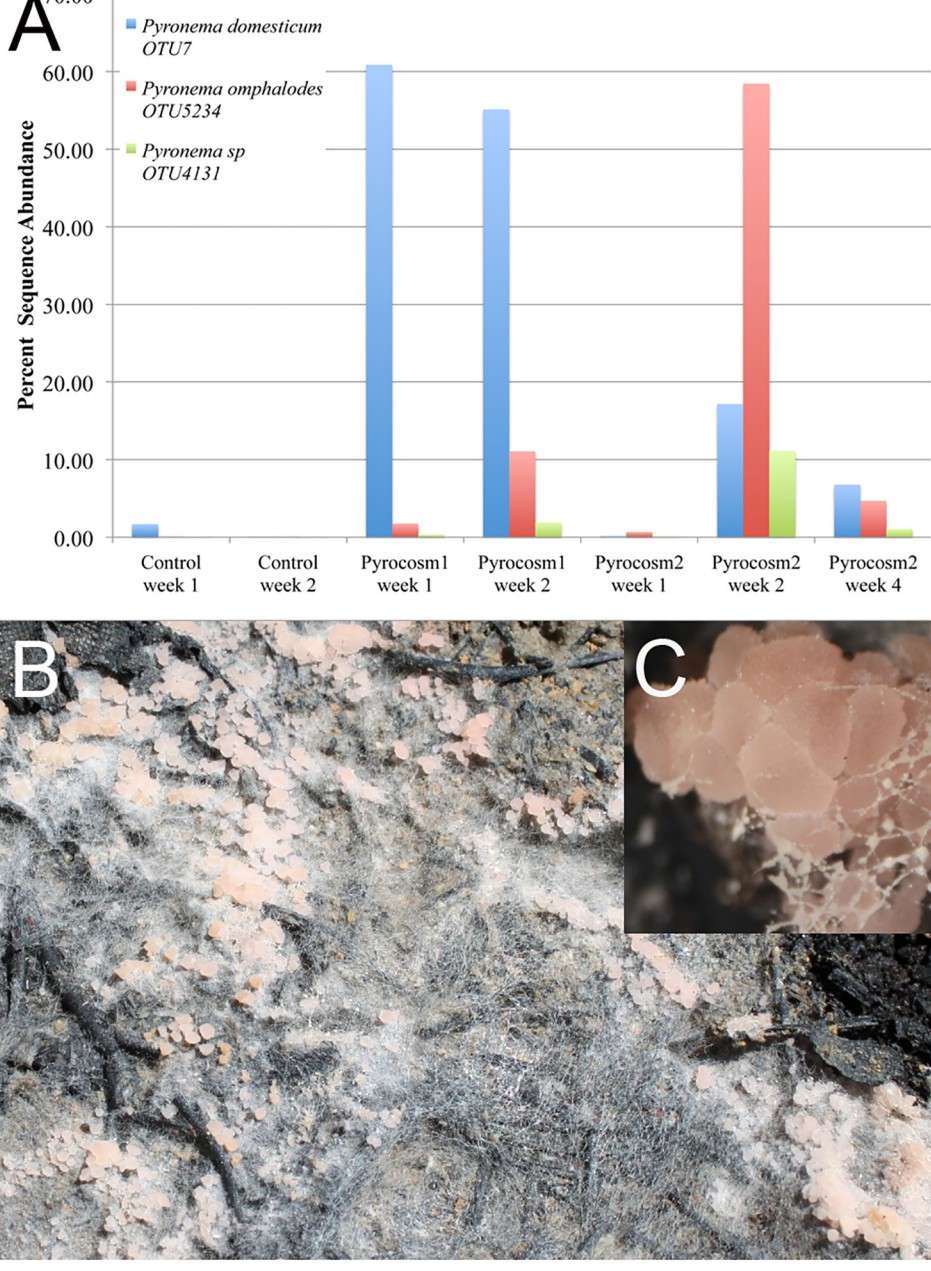

**Fig 3. *Pyronema* species react rapidly to simulated fire in pyrocosms.** A) post-fire percent sequence read abundance of three *Pyronema* OTUs is shown in the control, and Pyrocosm 1 in weeks 1 and 2 and in Pyrocosm 2 in weeks 1, 2, and 4; B) mycelium and apothecia (ascocarps) of *Pyromena domesticum* fruiting on the surface of Pyrocosm 2, 17 days after the fire; C) closeup of same.

relative to the unburned control in at least one of the pyrocosms, but only 10 OTUs showed increases in both pyrocosms relative to the control. Most of these increases were quite small, and only four OTUs increased by more than 1% relative abundance in both pyrocosms. Three of these were the *Pyronema* OTUs just discussed, the fourth was a *Russula* species (OTU73), an ectomycorrhizal fungus, that increased in sequence abundance by 1.37% and 3.7% in Pyrocosms 1 and 2, respectively. The remaining six OTUs that increased in both pyrocosms, did so

at much lower levels, and included two other ectomycorrhizal fungi (*Rhizopogon arctostaphyli*—OTU175, and an unknown *Thelephoraceae*-OTU879), two unidentified fungi (OTUs 304, 420), a Basidiomycota yeast (OTU631), and one additional *Pyronema* (OTU3739).

Evenness and richness were reduced in the burned pyrocosms relative to the control (Fig 4C), but composition differences, other than those related to *Pyronema*, were not striking (Table 2). The ranked abundance curves for the two experimental pyrocosms are very similar to each other, and distinct from that of the control (Fig 4). This pattern can also be seen in the

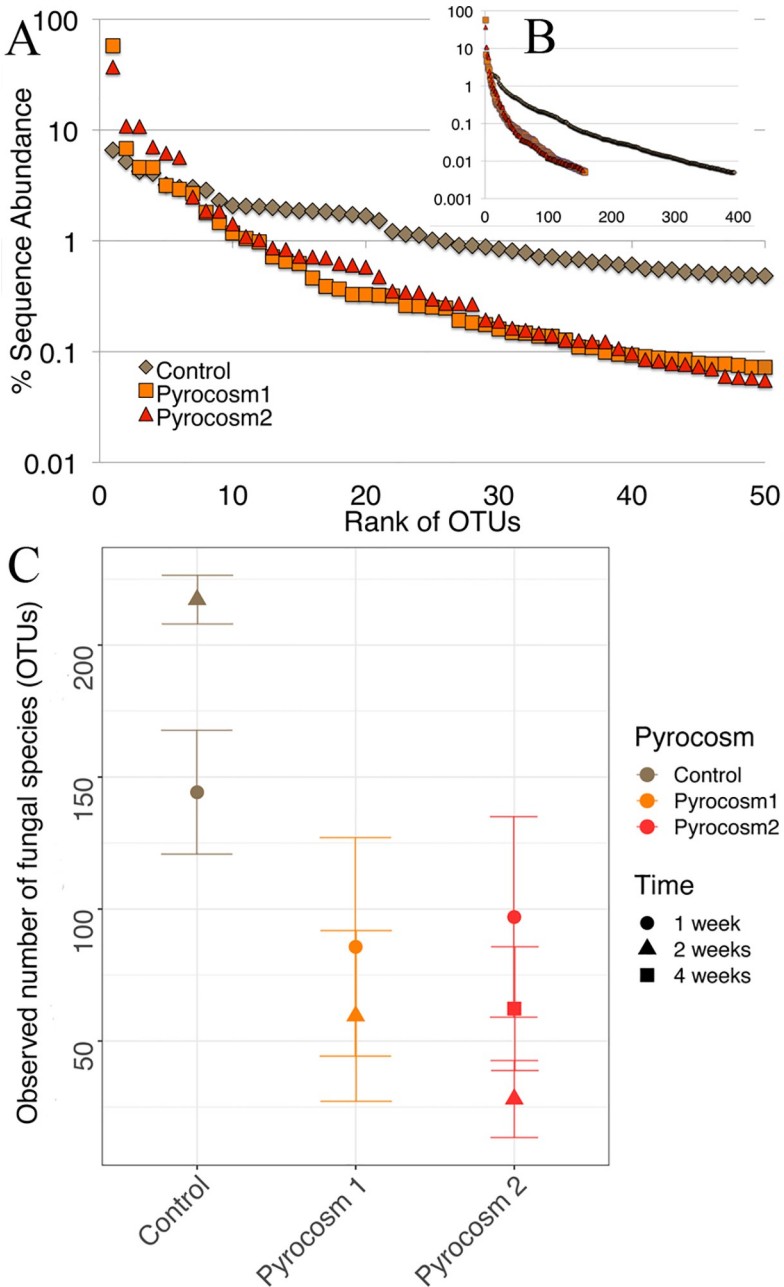

**Fig 4. Differences in diversity of fungal communities within pyrocosms and control.** A) Ranked abundance curves of pyrocosms and control based on % read abundance of for top 50 most abundant OTUs. B) Ranked abundance for all OTUs that are at least 0.01% of sequence. C) Richness estimates based on observed number of OTUs after rarifying samples to 3950 sequences; points show means for all samples; bars show SE.

level of dominance among component species. In unburned control the most abundant taxon, OTU 37 a xerotolerant yeast, represented 6.60% of the total sequence reads, and 25 OTUs each had more than 1% of the total reads. In contrast Pyrocosm 1 and 2 had only 11 and 12 OTUs, respectively, that each had more than 1% of the reads. Among the less abundant taxa, the control pyrocosm had 394 OTUs that were each represented by at least 0.01% of the sequence, versus 158 and 155 OTUs in Pyrocosms 1 and 2 respectively (Table 1, Fig 4).

The four no DNA controls revealed low levels of contamination as is typical of high throughput sequence studies. Of the 10 OTUs that were based on more than 5 sequences in these controls, 6 were identified as *Morteriella sp.*(Mucoromycota), four were identified as an unknown Ascomycota in the Dothideomycetes or Sordariomycetes, one was a *Cortinarius* sp. (Basidiomycota), and one was an unidentified fungus. Three of the *Morteriella* OTUs (64, 108, 794), and the unidentified fungal OTU (544), were found at similar or greater levels in the no DNA controls compared to the experiment. Of the remaining contaminating OTUs, three occurred at much lower levels compared to the experimental samples (OTUs 1,5,14) and three were found in the experimental samples at levels below the thresholds used for analyses (i.e., 0.01%, or 5 sequences) (OTUs 38,556,1021). None of these contaminants are relevant to the results or analyses discussed above.

All of the 10 knowns were recovered from the mock community controls, however, they yielded 16 OTUs with greater than 5 sequences each, and 20 OTUs with greater than 0.01% read abundance. This represents a 1.6 or 2-fold OTU inflation for these respective thresholds. This inflation was primarily due to minor amplicon variants of the known fungal species added, but also included some index switching from the experimental samples. The most obvious case of the latter involved *Pyronema domesticum* (OTU7), which was the most abundant sequence in the experimental samples (290,128 total reads), and it accounted for 0.22% (92 reads) in the mock community control, even though it was not included in the community.

## Discussion

We have shown that these very simple pyrocosms provide an experimental system in which soil heating can be achieved in predictable ways by simply altering course fuel levels. Based on prior soil heating models [11–13], we would expect soils that differ in heat capacities and water contents to have different slopes (i.e. Fig 2B). However, the predictability within a given soil type should remain high, and we would expect the lognormal relationship between temperature and depth to remain. The strong correlation of peak temperature with natural log of depth (Fig 2B), means that one can accurately predict peak temperatures at any depth if temperatures at two depths are measured, although the edge effects will broaden the range of temperatures achieved within a give depth zone. Earlier models have shown that water content of the soil will have a large effect on soil heating, particularly as water content rises above 8% [13]. High water contents were specifically avoided in our studies because we assume that most large wildfires are likely to occur during very dry periods. We have also found that dry soil performed quite similarly to our sand pyrocosms, which helps with predicting peak temperatures in novel soils. However, if one were to model soil conditions of prescribed burns, higher water content of soils should be considered.

These results, and those of earlier modeling studies, have revealed features of the physical environment that are likely to be important for survival of propagules in the soil. The high heat capacity of soil increases with temperature [11], and means that soil heats slowly but retains heat for a long time. This is shown by the fact that peak temperatures at depth in the pyrocosms are achieved hours after the fire has gone out (Fig 2A). A study by Smith et al [31] found the same pattern of lingering soil temperatures under large fuel load in intact forest

soils. Essentially the heated surface soils store heat and become the source for heating deeper soils. This means that at depths below a few centimeters, soil organisms would be "slow-cooked", with temperatures hovering around the peak for tens of minutes. This is important because the spore germination of *Pyronema domesticum* and some other pyrophilous decrease as heat treatments persist for several minutes [7].

Our results show that coarse fuels (e.g. charcoal) transfer more heat to the surface and ultimately to the deeper soils and were the best predictors of peak temperatures (Fig 2C). To relate this to a forest fire setting we envision a pattern of spotty heating across the landscape driven by the uneven distribution of coarse fuels. This view is concordant with that of Smith et. al. [31], who showed that log piles generate much higher subsurface heating than that achieved from the finer fuels in the surrounding litter. In highly intense crown fires, radiant heating of the surface soils are likely to contribute to much more soil heating than the litter [32, 33] and would likely even out some of this patchiness. Nevertheless, the uneven distribution of larger course fuels is likely to contribute to heating heterogeneity. The less homogeneous nature of *in situ* forest soil compared to the sieved, dried, uniformity of the pyrocosm soil would add an additional source of spatial variation in peak soil temperatures [34].

To visualize the changes in peak temperatures caused by fuel differences and relate them back to soil organisms, we have used our results to model the depth and width of the Goldilocks and necromass zones (Fig 5). For the sake of argument we will assume the Goldilocks zone exists between 50 and 70 ˚C. This temperature range was selected because lethal temperature for many organisms falls within this range [14], but spores of some fire adapted fungi are stimulated to germinate at these temperatures [9, 35, 36]. Similarly, we define the temperature limits of the necromass zone from 70 to 200 ˚C, a temperature range that is lethal to almost all organisms, but is not hot enough to pyrolyzed carbon substrates. This should therefore be a zone that is rich in labile carbon and nitrogen sources that are derived from heat-killed organisms.

Our model shows that the depth of the Goldilocks zone increases with fuel load (Fig 5A), while the thickness of the zone increases rapidly with addition of charcoal up to a threshold. After this threshold is reached, the width of the zone decreases slowly with additional fuel, but retains a similar width (~3–5 cm, Fig 5B) over most of the modeled range. Note that the actual temperatures that define the Goldilocks zone would likely vary with specific organisms [14], and might be hotter or cooler than assumed here. Nevertheless, different defining temperatures would only cause the zone to move up or down in depth and modify the width, but the basic pattern of a thin selective zone that varies with local fuel load would still be valid. When we modeled the depth and width of the necromass zone in relation to coarse fuel, we found a similar pattern (Fig 5). The width of the zone increases rapidly up to a threshold fuel load, and then stays relatively constant (Fig 5B). Therefore our model predicts that in a forest setting there would almost always be a survivable zone for pyrophilous microbes as long as the soil is deep enough, and a larger open niche (i.e., the necromass zone) would occur directly above this zone when higher fuel loads occur.

Soils above the necromass zone would be expected to have unique post-fire soil chemistry that might provide substrates for fire adapted species. Soil temperatures in higher than 175 to 200 ˚C volatilize waxes and lipids that then condense at lower temperatures in the soil below and cause a hydrophobic layer that can facilitate erosion [37]. This should occur near the top of necromass zone. It is not known if these concentrated hydrophobic materials serve as energy-rich carbon sources for microbes, but it is known that the hydrophobic layer disappear with time [37], and we think it would be surprising if microbes were not involved in some way. Temperatures greater than 350 ˚C convert cellulose and lignin into highly aromatic forms of biochar [16] that are relatively recalcitrant to microbial degradation [38]. According to our

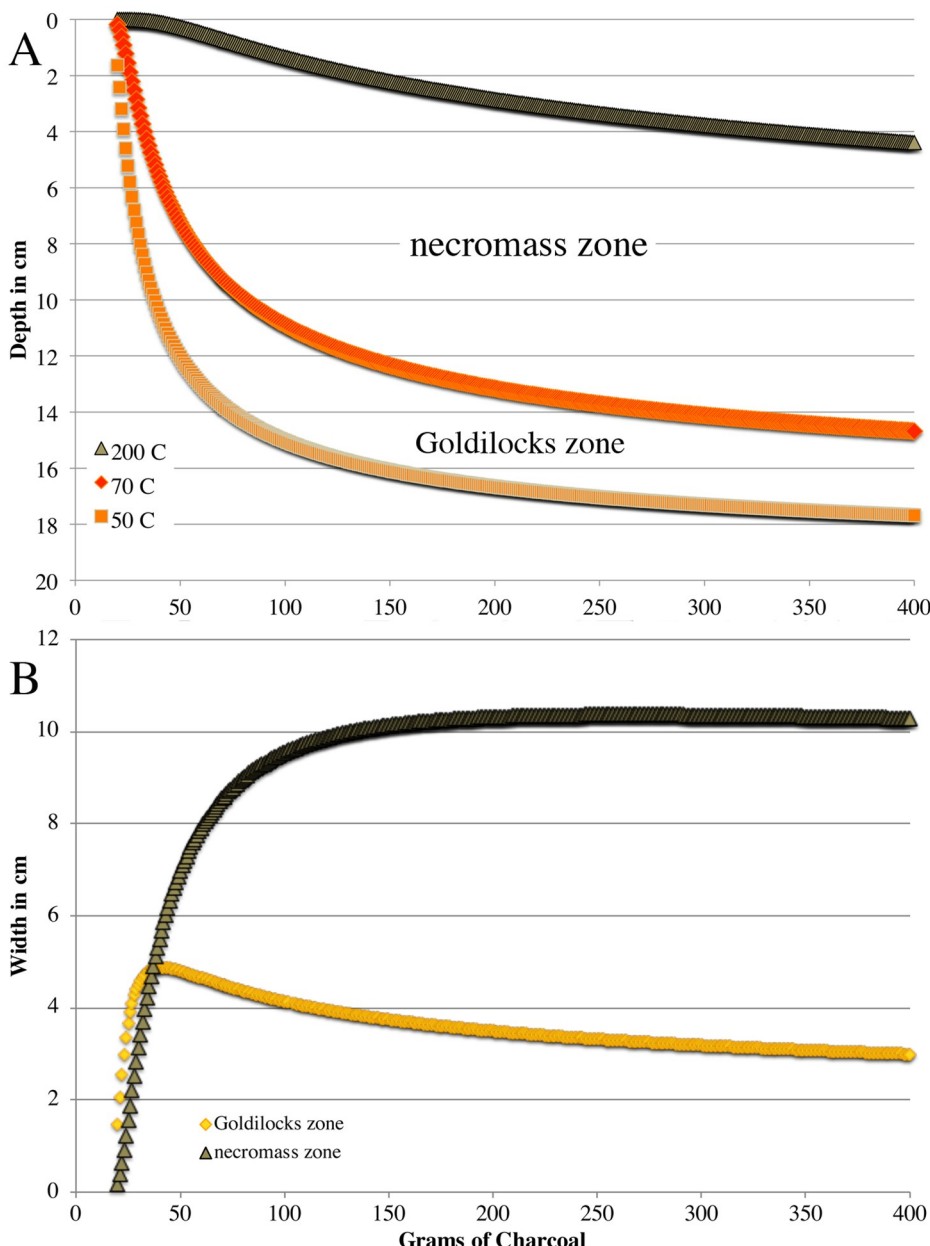

**Fig 5. Modeled behavior of the "Goldilocks" and necromass zones as mass of charcoal is increased. A)** depth at which peak temperatures reach 50, 70, and 200 ˚C are modeled by using the ln relationships between mass of charcoal and peak temperature at two given depths (10.5 and 15.8), and then predicting the depths of the selected peak temperatures (i.e. 50, 70, 200) using a ln relationship between depth and peak temperatures (i.e., Fig 2B). The Goldilock and necromass zones are labeled between the hypothetical temperatures that define them **B)** Width of both zones over the same range of charcoal mass.

regression-based model these temperatures would only occur within about a centimeter of the surface in the highest fuel loads tested, and would be non-existent in the lower fuel loads tested.

The rapid, massive response of *Pyronema* species in both fire pyrocosms (Fig 3) demonstrates that this system is capable of simulating at least some known pyrophilous microbes, and it reproduces a response of similar magnitude reported from prescribed burning of log piles [20]. *Pyronema*'s response in our system is impressive in that three OTUs accounted for

68.7 to 86.7% of all fungal sequence reads in both pyrocosms within just two weeks of the fire (Fig 3), and the most dominant *Pyromena* OTU in each pyrocosm accounted for 55.1 to 58.4% of all reads in the two-week post-fire period, compared to less than 1% in the control (Table 2). For comparison the most dominant fungus in the unburned control was a xerotolerant yeast that accounted for less than 7% of the sequence (Table 2). This latter value is typical of dominants in other studies of soil fungi in which the most abundant species typically account for only few percent of the sequences, and all species with more than 0.1% are viewed as dominants [39]. *Pyronema* species achieved a dominance that is roughly 6 to 10-fold higher than is typical of the most dominant fungal soil fungi, and they did so in just one or two weeks.

*Pyromena* species have been known to be rapid responders to fire for over a century [4], but all of the earlier literature is based on the fruiting response of the species. Evidence of its *in situ* mycelial response was lacking until recently [20]. *In vitro* studies have shown that *Pyronema* species grow rapidly in situations where there are few competitors [7, 40]. *Pyronema* was also mentioned in Petersen's (1970) experimental fire work, but its fruiting, though rapid, was not consistent enough for it to be considered in the successional patterns that Petersen discussed. Interestingly in our study, it only fruited in one of the two pyrocosms, even though it dominated in both. Thus, it may be much more common in post-fire soils then its fruiting record suggests. At least some pyrophilous fungi appear to have dormant spores or sclerotia in the soil, and these germinate following heating or chemical changes associated with fire [6, 15]. This mode of activating quiescent propagules is consistent with the rapid response we saw, but we did not specifically test for it by preventing dispersal. Nevertheless, the presence of all three dominant *Pyronema* OTUs in both pyrocosms shows that no inoculation was necessary for a rapid response.

The small response of *Pyromena domesticum* in week one in the unburned control is interesting (Fig 3), and corresponded to a total of 1.69% (3535 total reads) of the sequence in that sample. This is three orders of magnitude higher that the 92 sequence reads that contaminated the mock community, and therefore must be a real response of *Pyronema* within the unburned control soil. We interpret this as a short-term simulation following the wetting of the dried soil. One could imagine that such transient responses could take advantage of brief periods of low competition to renew soil inoculum between fire events. In any case, the prevalence of *Pyromena domesticum* in the control dropped to 0.006% by week two. The burned pyrocosms also show rapid increases and decreases in *Pyronema* within one-week periods (Fig 3). This rapid turnover shows that most of the *Pyronema* DNA in the soil does not linger long, and it suggests the occurrence of either autolysis or degradation by other components of the microbial community. In contrast the presence of ectomycorrhizal taxa such as *Cortinarius* and *Russula spp.* that occur in similar levels within the control and burned pyrocosms (Table 2) are likely due to survival of environmental DNA, perhaps as spores, as neither taxon would be expected to grow without a host.

It is less clear if fungi other than *Pyronema* specifically responded to the experimental fire in a positive way because abundance differences were not as great and replication was insufficient in this pilot study. However, the inclusion of ectomycorrhizal fungi in the set of fungi that appeared to increase after the fire at low levels (Table 2) shows that these apparent changes are within the level of stochastic sampling error, as growth of these ectomycorrhizal fungi could not have occurred without a host tree. For most other fungi we have no knowledge of their autecologies, so when they appear to respond at similar levels to the ectomycorrhizal fungi there is no strong evidence for or against their fire response. Longer incubations times and increased replication are necessary to resolve this.

The negative effects of fire on most fungi is clear from the reduced species richness and shape of the ranked abundance curves (Fig 4). This result is concordant with a meta-analysis of fungal response to fire [3], and makes intuitive sense based on the killing effect of soil

heating. However, having taxa with extremely high read abundance (e.g. *Pyronema*) can distort perception of community structure because our ability to detect less abundant taxa is reduced as more of the sequence depth is consumed by the dominants [41]. However, if we remove the sequence reads from the top three *Pyronema* OTUs, and recalculate percent dominance, the rank abundance curves from the burned pyrocosms are still much less even than that of the control (S3 Fig). This shows that in addition to *Pyronema* other taxa are also driving the shape of the ranked abundance curves, and thus the reduction in richness and evenness are unlikely to be artifacts.

We propose that the pyrocosm system is an excellent model system to dissect the post-fire microbial community, and this can be done in a variety of ways. For example soil source, fire intensity, watering regime, access to external inoculum and incubation conditions can all be varied and replicated to study the process of post-fire community assembly. If pyrocosm experiments were incubated longer or under different conditions (e.g., water, soil type, temperature), other common, known pyrophilous fungi [6] might develop from the natural inoculum just as *Pyronema* did. However, even if they did not, most of them grow well in culture [7] and could be added into post-fire soils that lack them in varying orders and combinations to determine environmental versus biological interactions that underlie community assembly [42]. There are also indications of parallel postfire responses in bacteria and microfauna [2] that could be studied in similar ways with this system.

Finally, we propose that *Pyronema* is an excellent model organism for the study of fire fungal ecology. In addition to their ease of isolation and rapid growth rates, three genomes from *Pyronema* species have now been completely sequenced, assembled and annotated, and are available on Mycocosm (http://jgi.doe.gov/fungi), the genome portal for the U.S. Department of Energy (DOE) Joint Genome Institute (JGI) [43;44]. Coupling these genomic resources with the rapid dominance of *Pyronema* in postfire soil can thus be used to explore the functional roles of this fungus in a realistic environment. For example, gene expression could be studied by incubating *Pyronema* on temperature defined soil layers removed from pyrocosms (S1 File) followed by extraction and sequencing of mRNA. This approach would help improve our understanding of what these fungi live on within the post-fire soil.

Furthermore, *Pyronema* is not unique within the post-fire community; genomes of 10 other pyrophilous fungi have now been sequenced, assembled, and annotated (Table 3). The list includes representatives of most of the common genera of pyrophilous fungi [6]. This is important because metatranscriptomic approaches with fungi are generally limited by the

**Table 3. Sequenced, assembled and annotated Genomes of pyrophilous fungi.**

| Phylum | Species | Website |
|---|---|---|
| Asco | *Geopyxis carbonaria* | https://genome.jgi.doe.gov/Geocar1/Geocar1.home.html |
| Asco | *Morchella snyderi* | https://genome.jgi.doe.gov/Morsny1/Morsny1.home.html |
| Asco | *Peziza echinispora* | https://genome.jgi.doe.gov/Pezech1/Pezech1.home.html |
| Asco | *Pyronema omphalodes* | https://genome.jgi.doe.gov/Pyrom1/Pyrom1.info.html |
| Asco | *Pyronema domesticum* | https://genome.jgi.doe.gov/Pyrdom1/Pyrdom1.home.html |
| Asco | *Tricharina praecox* (DOB1048*)* | https://genome.jgi.doe.gov/Pezsub1/Pezsub1.info.html |
| Asco | *Tricharina praecox* (DOB2270) | https://genome.jgi.doe.gov/Tripra1/Tripra1.home.html |
| Asco | *Wilcoxina mikolae* | https://genome.jgi.doe.gov/Wilmi1/Wilmi1.home.html |
| Basidio | *Coprinellus angulatus* | https://genome.jgi.doe.gov/Copang1/Copang1.home.html |
| Basidio | *Crassisporium funariophilum* | https://genome.jgi.doe.gov/Crafun1/Crafun1.home.html |
| Basidio | *Lyophilum atratum* | https://genome.jgi.doe.gov/Lyoat1/Lyoat1.info.html |
| Basidio | *Pholiota molesta* | https://genome.jgi.doe.gov/Phohig1/Phohig1.home.html |

number and taxonomic coverage of sequenced and annotated genomes [45]. Due to its relative simplicity, the post-fire soil fungal community may now have the best genomic resources of any soil system, and these resources can be used in combination with the pyrocosms to dissect interacts with post-fire soil chemistry or with other organisms in this environment.

## Conclusions

Pyrocosms now add to the growing list of experimental systems that can be used to study fungal community ecology [46], and they specifically relate to a natural environment that is growing in importance as fire size and intensity continue to rise [47]. Furthermore the community and the factors affecting this environment are simple enough that experimental and modeling approaches might be easily combined to achieve greater predictability [48], while the availability of genomic resources open avenues to dissect the functional roles of the component species in great detail.

## Supporting information

**S1 Fig. Effect of flash fuels is subtle.** 172 gm of flash fuels caused temperatures to rise slightly at 10.5 cm below the surface, but ultimately achieved the same peak temperature as solar heating in an unburned pyrocosm monitored simultaneously. Pyrocosms with the same flash fuel load, but one or two charcoal briquettes (25–51 g) heated more rapidly and achieve higher peak temperatures.
(TIF)

**S2 Fig. NMDS plots using Bray-Curtis dissimilarity.** Samples were rarified to 3950 reads/sample. Adonis $R^2$ = 0.33 and 0.19 for Bray-Curtis (A) or Jaccard (B) respectively.
(TIF)

**S3 Fig. Ranked abundance curves with the three most abundant *Pyronema* OTUs dropped and percent read abundance recalculated without them.** Top 50 most abundant OTUs are shown.
(TIF)

**S1 File. Additional notes on pyrocosm assembly and use.**
(DOCX)

## Acknowledgments

Genome sequencing of pyrophilous fungi was accomplished through a community sequencing project from the Joint Genome Institute to TDB.

## Author Contributions

**Conceptualization:** Thomas D. Bruns.

**Data curation:** Thomas D. Bruns, Sydney I. Glassman.

**Formal analysis:** Thomas D. Bruns, Sydney I. Glassman.

**Funding acquisition:** Thomas D. Bruns.

**Methodology:** Thomas D. Bruns, Judy A. Chung, Akiko A. Carver, Sydney I. Glassman.

**Writing – original draft:** Thomas D. Bruns, Sydney I. Glassman.

**Writing – review & editing:** Thomas D. Bruns, Sydney I. Glassman.

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
