## [Decision Letter · Decision Letter 0]

24 Oct 2019

PONE-D-19-24954

A simple pyrocosm for studying soil microbial response to fire reveals a rapid, massive response by Pyronema species

PLOS ONE

Dear Dr. Bruns,

Thank you for submitting your manuscript to PLOS ONE. After careful consideration, we feel that it has merit but does not fully meet PLOS ONE’s publication criteria as it currently stands. Therefore, we invite you to submit a revised version of the manuscript that addresses the points raised during the review process.

We would appreciate receiving your revised manuscript by Dec 08 2019 11:59PM. To enhance the reproducibility of your results, we recommend that if applicable you deposit your laboratory protocols in protocols.io, where a protocol can be assigned its own identifier (DOI) such that it can be cited independently in the future. For instructions see: http://journals.plos.org/plosone/s/submission-guidelines#loc-laboratory-protocols

We look forward to receiving your revised manuscript.

Kind regards,

Garret Suen, Ph.D.

Academic Editor

PLOS ONE

**Journal Requirements:**

2. Thank you for stating that “The funders had no role in study design, data collection and analysis, decision to publish, or preparation of the manuscript” in your financial disclosure.

Please also provide the name of the funders of this study (as well as grant numbers if available) in your financial disclosure statement.

**Comments to the Author**

1. Is the manuscript technically sound, and do the data support the conclusions?

Reviewer #1: No

Reviewer #2: Partly

Reviewer #3: Partly

2. Has the statistical analysis been performed appropriately and rigorously? 

Reviewer #1: No

Reviewer #2: No

Reviewer #3: Yes

3. Have the authors made all data underlying the findings in their manuscript fully available?

Reviewer #1: Yes

Reviewer #2: Yes

Reviewer #3: Yes

4. Is the manuscript presented in an intelligible fashion and written in standard English?

Reviewer #1: No

Reviewer #2: Yes

Reviewer #3: Yes

5. Review Comments to the Author

Reviewer #1: This manuscript was problematic for me. One the one hand, it clearly shows a methodology that could be used to study the effects of soil heating (by fire) on soil microbial communities. In the regard it is useful. On the other hand, the manuscript did not adequately test the proposed methodology. We are not shown how much variability occurs among multiple replicates, we do not know whether the conditions the system is able to produce in the soil actually mimic the conditions during a fire in the field, we are not shown how to alter fuel loads, etc. to tailor conditions to mimic a variety of field conditions during fire in the field, etc. etc. Therefore, this contribution is not nearly as useful as it might be. On balance, therefore, I think the manuscript could be publishable if the authors could remedy the deficiencies I have pointed out.

The introduction does a great job of introducing the reader to the effects of fire on soil chemistry, physics and biology. But there is little justification for the development of a “pyrocosm” system for experimentally producing heated soils. Are there no other systems that have been previously developed and used for similar purposes?

The manuscript appears to be written hastily (see below).

Line 15. There is one too many “the”s in the sentence.

Line 17. This sentence indicates that “peak soil temperature” lingers near peak temperature, which is probably not what was intended.

Line 20. Should read “of charcoal”

Line 22. Is “massive” the best descriptor?

Line 25. How is it possible for there to be a reduction in richness while the other fungal taxa did not change significantly?

Line 29. Not clear what “it” refers to.

Line 33 uses the term “pyrophilous fungi”, while line 23 uses the term “postfire fungus”. Are these the same? If so, please use a single term.

Line 81. If they are “partially burned”, they can only be “partially pyrolyzed”, not “pyrolyzed”.

Line 81. Heat alone is not sufficient for pyrolysis. Oxygen must also be absent or very low.

Line 115. The written description here is inadequate. The photo helps, of course, but the written description ought to be accurate. “Filled to depth of 16 cm” does not tell how much space occurs above the soil within the bucket. And what does “buried to soil level” refer to, the soil in the bucket or the soil in the field? Which soil surface (in or outside the bucket) were the 10-13 holes positioned above?

Line 158. What is meant by the “first” soil pyrocosms? Were there more than one? How many? It appears that there were only two pyrocosms total ever used for testing. If that is true, the obvious difficulty is that we do not know how much variability exists among pyrocosms. If you only use 2, it is possible that by chance they were similar.

Line 168. Why 10.5 and 15.8 cm? Is that arbitrary?

Line 176. How were the samples dried?

Line 177. When tamping down, were field densities approximated?

Line 180-185. Why were charcoal briquettes used? Does their heat simulate that of burning wood? Is heating to a certain temperature in x amount of time equivalent to heating to the same temperature in 2x amount of time? Or is that unimportant?

Line 207. Again, how does a single replicate serve as a control?

Line 274. I do not think one can say much about reproducibility from n=2. True, the two look very much the same, but that could occur simply by chance. What if there were 4 or 5 or 10 replicates?

Line 275. The fact that peak temps are reached hours after the fire is out and linter for 40 minutes seems to be an artifact of the conditions. If there were a hotter fire, if there were thicker soil, if the buckets were a different size, etc. the results would be different. So how is it possible to know whether these conditions are relevant to those that occur in natural fires in the field?

Reviewer #2: The contribution describes a simple pyrocosm, in which soil communities and/or processes can be inexpensively studied in well replicated experimental settings. The value in this paper is not in its data or robust statistical analyses, but rather in the design of the experimental “fire bucket” and in a series of experiments that serve as a proof of concept for the experimental system. The manuscript demonstrates that the heat transfer into the soil profile - at least with the soils used - can be predicted by soil depth and mass of charcoal used in the experiments. Further, the acquired data suggest that pyrophilic organisms seem to respond in fire stimulation in the experimental system as inferred from MiSeq sequencing of the soil samples from the pyrocosm.

I see the value of describing and marketing this system as a valuable platform for fire studies without the excess complications from environmental noise. However, the absent replication minimizes the true inference of the organismal responses, and should be explicitly – and more concisely – be used as further proof for the system’s functionality. The meandering discussion on the poorly supported organismal responses is excessively lengthy and should be dramatically shortened. On the topic of presentation, the word microbes is used frequently, but in the body of the paper only fungi are discussed – are fungi microbes?

Also, the modeling of the potentially biologically interesting depths in the system is interesting but is described in excessive lengths. Although I do enjoy the modeling of the “Goldi Lock” zone, I think this could be more briefly described in the text and much of the detail moved to supplements.

I have a number of minor concerns that I list below.

MINOR COMMENTS:

Line 40: serotinous

Line 11: how is it possible to fit 7 liters of soil into a bucker that holds 3.8 liters?

Line 123: had

Line 128: is flash fuel tooth picks or pine needle litter? Compare to line 121

Line 135: Clarify what you mean by soils were collected separately

Line 196: I am uncertain how well does the single briquette transfer heat to a bucket of this size. The heat transfer is unlikely linear through the depth

Line 213: What is the depth of the core sample? What are the four even zones, based on depth?

Line 227: specify log (10, e, or something else)

Lines 301-303: I am not sure if the modeling correctly applies to flash fuels. I argue that heat transfer is a function of both duration and the temperature of the heat pulse. Therefore, extrapolating the models to flash fuels may be erroneous.

Line 315: effects

Line 15 on pager 21: Pyronema

Line 20 on page 21: It is curious that Pyronema would appear in the mocks. I have two questions: 1) were there any other taxa in the mocks; 2) is it possible that the mock is contaminated, because it was prepared in a lab that may maintain Pyronema cultures?

Lines 33-39 on page 22: While I do not question whether or not Pyronema spp. were stimulated by the fire treatments in pyrocosms, I am curious about the ectomycorrhizal fungi. I would ask the authors to elaborate on this - perhaps in the discussion. I would be particularly interested in discussion that would consider relative abundance increases when overall sequence pool changes (lower richness) and taxa (or their nucleic acid traces) may have been lost as a result of the fire treatment.

Line 69 page 23: why were the data rarified to 3950 sequences. There were a total of 6000 sequences for each of retained samples

Lines 121-122, Page 26: I believe this to be context dependent. While true for low severity fires, in wildfires that include whole log combustion the high temperatures last longer and heat is transferred deeper. See Smith et al (2016 - International Journal of Wildland Fire 25(11) 1202-1207).

Line 109, 110, 142 – In these lines, it was discussed that the soil was dried to mimic dry soil that would probably occur during a wildfire. Crushed ice was added the day after fire and the following days after to mimic snow fall. How realistic is this to dump ice onto a freshly burned plot to mimic slow snowfall? Would snowfall occur that close in time after the fire.

Line 176 Page 28: It is odd to return to definition of the necromass zone at this point. Perhaps the discussion points could be mane more concisely.

Line 202, Page 29: it may be an over statement that there was no prior evidence of mycelial in situ response for Pyronema. If my memory serves, Reazin et al (2016 - Forest Ecology and Management 377:118-227) included Pyronema observations.

Line 448: Please include the NCBI BioSample and representative sequence accession numbers.

Reviewer #3: Review of PONE-D-19-24954

This manuscript details a novel method of studying the response of fungi and other microbes to fire through the use of ‘pyrocosms.’ These in situ systems can be used to closely monitor the thermal dynamics of fire-frequented systems and study the effects of fire on microbes through inoculation. The authors use pyrocosms to describe temperature dynamics following fire in detail, and to stimulate the growth of native, pyrophilous fungi (Pyromena). Construction of the pyrocosms is well-documented. Given that the pyrocosms are cheap, easily reproducible, and can be used in a wide variety of applications, this work is a useful addition to the literature and suitable for publication in PLOS ONE.

The strengths of this paper are x and y, In revealing thermal dynamics in close detail and stimulating native, pyrophilous fungi, the authors demonstrate the clear utility of the pyrocosms, but the writing inhibits communication of this topic. For instance, I still have questions about how the experiments with the pyrocosms were conducted. So that readers can better visualize how they were used in these two experiments and in their own research, I recommend more thoroughly reporting how your two hypotheses/goals (thermal characteristics of the system and stimulating fungi) were tested in the Methods (see specific suggestions below). This being said, I found File S1 very helpful in understanding how the pyrocosms were created. In addition, if it is a key goal of the manuscript, your discussion of how a ‘Goldilocks zone’ was determined should be detailed here (along with a more explicit discussion of it in the Intro and Results) in place of solely the Discussion. I feel cautious about the number of replicate pyrocosms used in both experiments. From the writing, it is unclear how many sand pyrocosms were used in the heating experiment. Further, there were only 2 pyrocosm replicates for ‘soil’ portion of this study, and these 2 replicates contained different soils. With these 2 replicates, you can only soundly conclude that this system stimulated the growth of Pyromena. Because of this, there needs to be a more explicit statement that increased replication is needed to confirm the patterns of richness you observe, as well as an explanation for why you chose to use only 2 replicates in the Methods. Lastly, there are several potentially confusing wordings (see below for suggested copy edits). Clearing these confusions will ultimately create a tighter and more cohesive manuscript, and enable you to make a stronger case for the utility and applications of the pyrocosms.

Abstract/Introduction:

15 Should be “for studying”

27 Should be “wildfires”

37-51 Specify if you are considering wildfires only here, or fire in general (wildfires + low-intensity prescribed fires). While true of wildfires, some of these things, specifically high fungal mortality, can’t always be extended to prescribed fires.

39 Get rid of “either”

55 Consider changing “burnt” to “burned”

61-62 Although I recognize that we don’t know much about pyrophilous fungi in comparison to other groups, claiming that we don’t know what they are doing in post-fire environment seems slightly out-of-place, especially since you state earlier that we know they are saprotrophic.

64-71 From later portions of the manuscript, it does not seem like ectomycorrhizae are a central focus. Removing this paragraph would make for a better focus on your central goals.

95 Should be “wildfires”

100 This is not necessarily true for prescribed fires. Consider changing language to focus on wildfires.

100 should be “post-fire”

105-110 In reading the rest of the manuscript and from lines 90-92, I understand that you used the temperature experiment to investigate effects of fuel load and to determine a potential ‘Goldilocks zone.’ Mentioning these goals here will help readers track these ideas throughout the manuscript.

I assume that you aimed to use the pyrocosms in ways that mimic wildfires, rather than prescribed fires. However, for reader clarity, reiterate this when you state your hypotheses/goals. Also add that the pyrocosms directly answer to the need for increased sampling immediately following fire.

Methods:

126 Pictures are great for visualizing the pyrocosms!

132-146 Is there a specific reason why only 2 pyrocosms (and 1 per soil type treatment) were used to study fungal response? This is a very low number of replicates, and should be justified.

138 Why was this soil not depth stratified?

139 Confused about the litter and F layers.. these were collected from the unburned site, and then used in both pyrocosms?

149 How many sand pyrocosms were used? “14 experiments” seems vague.. does this mean pyrocosm replicates, number of times the pyrocosms were burned, or a combination of the two?

154 Here (and in line 167, and in Results) you imply that you used the sand pyrocosms to study thermal dynamics under different fuel loads. If this is the case, I would restate this more explicitly, as well as what the fuel loads were.

190 Does this mean that the soil pyrocosms were included in the temperature experiment?

193 For consistency, keep this in g only. In addition, why did the amount of charcoal added to the sand pyrocosms differ so much?

192-196 Why is this ignition protocol different from the one you describe for the soil?

203 Although I assume that the main point of adding ice to the pyrocosms was to stimulate fungal growth (mentioned line 123), it would helpful to remind readers why you chose to do this.

206 Why did the soil pyrocosms receive different amounts of ice?

208 So that readers do not get confused, move discussion of the 3rd forest soil pyrocosm (the control) to 132-146.

213-217 Does this mean that you took one soil core from each pyrocosm at each time point? Why was the 4-week sample taken only for the 2nd pyrocosm? Did you sample the entire depth of the pyrocosm (meaning 4in depth increments)?

221 Is there a specific reason why you chose to amplify ITS1 instead of ITS2 (ie is it more represented in UNITE)? If so, please cite here.

258 Provide full name for OTU (operational taxonomic unit)

Results

273-315 is this temperature data coming from the sand pyrocosms or the soil pyrocosms? The methods reflect that the sand pyrocosms were used for this. However, this section (especially the mention of 2 replicate pyrocosms, line 282, and Fig. 2A) makes it seem like this data is from the soil pyrocosms.

282 If this corresponds to pyrocosm 1 and 2, make sure that it reads this way in the figure (ie change A and B to 1 and 2).

292 Should be “0.94779 respectively”

295 To help readers better track ideas, the flash fuel/fuel load portion of the heating experiment should be mentioned in the Intro and Methods.

303 “to a depth of 1.26 cm”

310 “for the surface” is a little confusing

315 should be “effects”

Table 1: Should pyrocosm 2 week 3 be week 4 (see Methods line 214)?

320 “in control or treatment pyrocosms”

326 “Table 1”

Table 2: should be “Control”

5 (and on): capitalize pyrocosm 1 and 2 (ie “Pyrocosm 1”) to keep consistent with Methods

10 “ranked 10th the 4th in abundance” is confusing.

Discussion

141 If a key aspect of your study is finding exploring heating patterns and Goldilocks zones, it should also be discussed in the Intro, Methods, and Results.

173 What disappears with time? You might be missing a word here

246-259 Your statement that more replication is needed needs should be stronger and repeated in this paragraph. As mentioned above, only having 2 replicates calls these conclusions on richness into question.

274-294 This whole-genome sequencing section might be better combined in a sentence or 2 with the benefits of Pyromena as good natural inoculum. I believe that your manuscript has a nice focus on the benefits of pyrcosms and how they can stimulate pyrophilous fungi, and the level of detail on this sequencing portion is not necessarily needed.

297-303 I might add that they answer to the need for increased sampling immediately following wildfire.

6. PLOS authors have the option to publish the peer review history of their article (what does this mean?). If published, this will include your full peer review and any attached files.

Reviewer #1: No

Reviewer #2: No

Reviewer #3: No

---

## [Author Response · Author response to Decision Letter 0]

23 Jan 2020

Here is the amended statement on funding that you requested: 

“The funders had no role in study design, data collection and analysis, decision to publish, or preparation of the manuscript. The work was funded by the Department of Energy grants DE-SC0016365 and DE-SC0020351 to TDB.”

And here is the updated data availability statement: 

The underlying data for the work consisting of: temperature and fuel data for all pyrocosms and the complete OTU table representing fungal community response. These are deposited in Dryad: doi:10.5061/dryad.45gd695; Representative sequences for all the OTUs are deposited in NCBI MN724033-MN724919, and the raw sequence read data are deposited in the short read archive: PRJNA559408. R-scripts for all sequence analyses are deposited in Github: https://github.com/sydneyg/Pyrocosms

General comments about the revision: We thank the reviewers for their careful reading of the manuscript and their many useful edits, suggestions and comments. They have certainly helped us refine our message and the presentation of results in this revision. 

 Comments on content are addressed below in the line-by-line list by reviewer. Small edits (spelling, grammar, word use changes) suggested have been made and can be found in the marked up version of the revision but will not be covered below. 

 The most general criticism made was about the limited replication (two experiments) in the biological part of the study. We concede that this limits the organismal inferences that can be drawn, and we had acknowledged this in various parts of the first draft. However, it serves the purpose of demonstrating that our simple experimental system works to stimulate at some pyrophilous fungi, and reviewer 2 and 3 recognized this, but thought it should be clarified. Similarly at least two reviewers commented our description/presentation of the “Goldilocks zone”, and though found it interesting, thought that it was not well incorporated or was overly discussed. 

 In response to these comments we clarified our goals and role of the biological experiments in the introduction. We did this by outlining our thermo-chemical gradient model in the third and fourth paragraphs. This succinctly summarizes the way we theorize that heat, soil depth and altered soil chemistry interact to create a predictable, depth-structured habitat for microbes in post-fire soils. In the last paragraph of introduction, we make it clear that: 1) our primary goal was to develop an experimental system that allows the post-fire soil system to be manipulated at a fine scale; 2) the forest soil experiments were essentially a pilot study to show that our experimental system has biological relevance, and 3) the heating results are used to refine our concepts of the Goldilocks and necromass zones in our model. 

Below we address the specific concerns of the three reviewers. 

Reviewer 1

Thanks for your comments and edits. However, we are not completely sure that we understand exactly what your concern about the lack of replication is referring to. If it is the biological replication, we concede that two reps and one control is not enough to generalize, but as discussed above we have now clarified that they serve primarily as a proof of concept. If concern is referring to the heating experiments then this is incorrect. As we showed with regressions that soil heating is highly predicable from our data. Specifically peak temperatures at difference depths are related by the ln of the depth (Fig2b), and peak soil temperature at a selected depth are a function of the mass of fuel (Fig 2C). The R2 on all these regressions is quite high, and replication of specific fuel loads is not necessary to demonstrate this. Nevertheless, we show that replicated fuel loads result in nearly identical heating profiles (Fig 2A). This can also be seen in Fig 2C for fuel load just under 300g. 

In terms of “showing readers how to alter fuel loads” we have made the methods clearer now, but we think that part of the problem may be that the reviewer apparently missed a supplementary file (S1) that goes into more detail on construction and uses of the pyrocosms. 

We have now reworked the introduction to better justify the development of the pyrocosms. This is specifically addressed in the last paragraph of the introduction. Thank you for this suggestion. 

Former line 22 - Yes, we think “massive” is an apt descriptor for a response that increases the sequence abundance of an organism from barely detectable levels to roughly 60% of all sequence. This is an order of magnitude more dominant than is typical of soil fungi in sequence studies. 

Former line 25 - The reason reduction in richness can occur without large changes to the other fungal taxa is because we use a relative abundance measure (i.e., sequence %). Thus large increases in a very small subset of taxa (e.g. Pyronema) decreases the apparent abundance of other taxa. The taxa “lost” were those that had very small abundance in the pretreatment condition, but taxa that were reasonably abundant are retained at similar levels. This is covered more in the results and discussion.

Former line 29 - now reworded to make the antecedent of “it” clear.

“Pyrolyzed” is now qualified were it needs to be and defined more carefully. The reviewer’s comment about the need for low oxygen for pyrolysis may be technically correct, but in practice this is clearly not a limitation in soil systems, or even soil surfaces, where charred remains are common. 

Former line 33 - Pyrophilous is now used uniformly

Former line 115 - We have revised the description of the pyrocosm setup to make it clearer and corrected the volume discrepancies that other reviewers caught as well. We also point to the supplemental notes (File S1) that have addition images and details. 

Former line 158 and elsewhere the number and type of pyrocosm experiments is now clarified. 

The selection of depths for the thermocouples was somewhat arbitrary but is explained on lines 310-313: “These depths were selected to insure that the wire thermocouples did not experience temperatures above 200ºC across the range of fuel loads tested.” 

Former line 322; Method of drying litter is now explained 

Former line 322 - F-layer and litter layer density was not measured; the tamping down was meant to approximate it, but we have no measure of how well we achieved that. However, it was completely incinerated, and we show that such flash fuels causes negligible soils heating, so we doubt it matters for the reproducibility or realism of the experiments.

Former line 180-185 The use of charcoal is now explained more fully. Essentially we use charcoal because they burn very predictably, transfer heat well to the soil, are probably not too different from natural substrates in forest fires, and are readily available. 

Former line 275 - The finding that peak temperatures at depth are achieved hours after the fire goes out and linger for ~40 minutes is absolutely not artefactual; it is caused by the large heat capacity of soil. This makes soil resistant to temperature change, but allows it to absorb and hold heat. Thus the heat source for soil at depth is the soil above it, not the active combustion. The result occurred in all the sand pyrocosms and both the soil pyrocosms, and it is likely to be greater in nature where the volume of soil is larger and there are not unheated edges. You can see evidence of this same effect in Smith et al paper cited where they burned log piles in nature. 

Reviewer 2 

The first comments about the value of this work being in the system, and the value of the sequencing being in the demonstration of its biological value is exactly what we had hoped to communicate. After reading the reviews and realizing that this was not obvious to everyone, we reworked the introduction as discussed above. Thank you for this clear insight. 

We agree that our limited biological replication limits inference. Our attempt to point out these limitations and to extract those results that transcend them was part of the reason the discussion was longer than the reviewer desired. We have now dropped at least two paragraphs of results and discussion about the biological response. What we retained was focused on Pyronema, and explaining why the mycorrhizal “response” indicates the noise level. 

The term “microbe” is not a taxonomic term - it’s simply a common name for small organisms. So yes, we see microfungi as a subset of microbes. Using this term particularly in the introduction is useful, because bacteria, and fungi both have to deal with the same fire and post-fire conditions and substrates, and they would both be assessed through very similar high-throughput sequence approaches. However, we have now tried to clarify exactly when we are referring to only fungi, which is obviously most of the time. 

We retained the modeling of the Goldilocks and necromass zones but as discussed above have now incorporated better into the focus of the paper as part of a broader thermo-chemical model. 

Minor issues

Former line 11 - Yes, we had given the wrong size for the buckets - nice catch.

Former line128 - flash fuel is now defined more clearly. 

Former line 135 - the soil separation is now spelled out better.

Comment on former line 196 - if you check the results (Fig S1) heat from one briquette was easily detectable, and had a greater effect than a much larger mass of flash fuels. 

Former line 213 - the depth of sampling is now specified (all the way to the bottom of the bucket)

Former line 227 - natural log is now specified but the truth is that log10 is almost as good. 

Comment on former lines 301-303: - I think there is confusion here on what is being regressed and we have now corrected that wording to clarify it. Basically there are there two main regressions used in this study. 1) temperature regression, which uses measured temperatures at 2 depths to predict temperatures at unrecorded depths (i.e, Fig 2b); and 2) course fuel related regression - where grams of charcoal are used to predict peak temperatures at particular depths (Fig 2C). We agree completely with the reviewer that flash fuels cannot be predicted by the second regression, but that is not what we did. Instead we are just using the measured peak temperatures achieved by the flash fuels at two depths to predict temperatures at other depths. This is valid because it is simply a function of the heat capacity of the soil - any heat source can be modeled this way. 

Former Line 20 page 21 - Other contamination sources are always possible with PCR, and that is why no-DNA and mock community controls are so important. What we show is that the contamination level is quite low. However, it is very common for mock communities or other control results to be contaminated by abundant sequences from other samples. This is because of “index switching” where the primer tags from one sample are recombined with the amplicon from another sample during PCR is common (see the Carlson et al 2012 paper cited). The more abundant the amplicon is, the more likely that it will experience index switching and end up looking like it was in samples it did not actually occur in. Thus, we think this is the most likely source of contamination. To avoid this, recent libraries have used double indexing (tags on both primers). 

Former lines 33-39 p 22. The nominal mycorrhizal “stimulation” was clearly artefactual, and that is (and was) mentioned in the discussion. We think that from the collective knowledge of Russula species we can say with some confidence that it did grow in a soil environment without a host. It is likely that the few percent increase seen in the two burned pyrocosms relative to the one control is due to soil heterogeneity/sampling issues that are compounded by low replication. We think the level of “mycorrhizal response" is a nice benchmark, in that we can’t infer anything about taxa that were apparently stimulated to similar or lower levels. 

Former line 69, page 23; 3950 was correct, not 6000. We corrected that now - nice catch!

Line 121-122 larger, glowing, heat sources certainly heat the soil more and result in higher temperatures at depth, but heat transfer follows the same physics even when the heating is less - that’s what our first regression shows. Even the slight heating from flash fuels showed the slow rise and the lingering effect (Fig S1); the main difference is that the peak temperature achieved is much lower/g of fuel. However, whatever peak temperature is achieved is will linger near that temperature for some time, and organisms need to endure it; that’s the point. 

Former Lines 109 etc, about ice: Water needs to be added. This is dry soil, so without it little activity would be expected. The various amounts and ways it could be added are clearly another variable that could be tested. The advantage of ice is that it melts slowly and wets the soil thoroughly. We were worried liquid water would channel through the hydrophobic upper layers and create uneven wetting, and would need to be added a little at time to avoid this problem. With a single pyrocosm that could be done, but with multiple pyrocosms the ice is much easier and provides a very uniformly timed wetting. Is a good mimic of snow? It’s clearly not perfect, but it’s probably good enough. The one big Sierra fire (Rim Fire 2013) we followed was initially wetted by a snow event that melted quickly at the elevation that these soils came from, and it spawned fungal activity similar to that we observed in the pyrocosms. 

Former line 176 - we have incorporated the necromass zone discuss into the paper better now (see bit on introduction), and reordered it in the discussion

Former line202 p 29. We had missed the Reazin et al. paper before, but have now incorporated it. It’s an excellent addition as it shows a massive Pyromena response in the field. We think this is the only such published example, although we have unpublished results from the Rim fire that show this too. 

All sequence samples now have accession numbers rather than the XXXXX seen in the original submission 

Reviewer 3

As discussed above we have now focused the paper on the pyrocosm system and the thermal-chemical gradient model, and this was partially in response to reviewers’ clear summary of the manuscript - we thank you for that. We have also clarified how many pyrocosms were involved and addressed the replication issues. 

Thank as well for the many edits suggested - these have been incorporated, but they will not be addressed line-by-line here. 

Former lines 61-62. Knowing that most pyrophilous fungi are saprobes really only scratches the surface in addressing the question of what they do in this chemically altered environment. We have now clarified our point by stating: “…, nor is it known what these fungi degrade and live on in the chemically altered post-fire environment.”

Former line 64-71 Discussion of ectomycorrhizal response to fire is now removed as suggested. 

Former line 100: The distinction between wildfires and prescribed fires is now clarified throughout. 

Former line 105 - The Goldilocks zone is now incorporated better. 

Former line 138 - the postfire soil added was not depth stratified because it was collected from a disturbed pile of soil in the fire zone. This is now mentioned. 

Former line 139 - the origins, use, and treatment of the litter and F-layer is clarified now.

Former line 149 - As explained in the set-up they were reused by removing ash following one experiment before adding fuel for the next. That is why we said “14 experiments” rather than 14 pyrocoms. We have clarified this now.

Former line 154 - Soil pyrocosms were also used for temperature experiments to the extent that the Ln relationship between depth and peak temperature are shown with soil (Fig 2B). This is indicated in the legend. 

Former lines 206 - The amounts of water differed because we had not standardized the amount of water yet. As explained above the biological side of experiment was a pilot study. If it had not succeeded in producing a biologically relevant result, the rest of the study would not have gone forward. So essentially we were trying different water and soil to see if anything worked. 

Former line 192-196 The soil experiments were actually run prior to the sand experiments. We only learned later that could create the same temperatures profiles with less manipulation by simply letting fewer charcoals burn completely. In the more detailed instructions on assembling pyrocosm (File S1) the difference between the two burning methods is discussed more, but the bottom line is that either works. 

Former line 203 - the ice is now explained more fully, and the explanation for the differences in ice are explained above. 

Former lines 213-217 - The coring number and depth is now clarified. 

Former lines 221 - our choice of ITS 1 was basically so we could compare it to our own earlier data. - This is now explained. 

Line 258 - Operational Taxonomic Unit had already been spelled out at first use of the OTU on line 228 (now line 376)

Line 273-315,and line 182. Temperature results are primarily from the sand pyrocosm, and the specifics are now indicated in the legend to Fig 2

Former lines 274-294 - The whole-genome sequencing section is germane to the system as a whole, because it makes it more experimentally more powerful. We spent a short paragraph on it, and we feel this is important so that others that might want to use the system are aware of the resources available.

---

## [Decision Letter · Decision Letter 1]

6 Feb 2020

PONE-D-19-24954R1

A simple pyrocosm for studying soil microbial response to fire reveals a rapid, massive response by Pyronema species

PLOS ONE

Dear Dr. Bruns,

Thank you for submitting your manuscript to PLOS ONE. After careful review, your manuscript has been deemed suitable for publication. However, the reviewers did catch some minor grammatical errors and points for clarification. I would ask that you correct these small errors and resubmit a revised manuscript so as to permit formal acceptance.

We would appreciate receiving your revised manuscript by Mar 22 2020 11:59PM. To enhance the reproducibility of your results, we recommend that if applicable you deposit your laboratory protocols in protocols.io, where a protocol can be assigned its own identifier (DOI) such that it can be cited independently in the future. For instructions see: http://journals.plos.org/plosone/s/submission-guidelines#loc-laboratory-protocols

We look forward to receiving your revised manuscript.

Kind regards,

Garret Suen, Ph.D.

Academic Editor

PLOS ONE

Reviewers' comments:

Reviewer's Responses to Questions

**Comments to the Author**

1. If the authors have adequately addressed your comments raised in a previous round of review and you feel that this manuscript is now acceptable for publication, you may indicate that here to bypass the “Comments to the Author” section, enter your conflict of interest statement in the “Confidential to Editor” section, and submit your "Accept" recommendation.

Reviewer #1: (No Response)

Reviewer #2: All comments have been addressed

Reviewer #3: (No Response)

2. Is the manuscript technically sound, and do the data support the conclusions?

Reviewer #1: Yes

Reviewer #2: Yes

Reviewer #3: Yes

3. Has the statistical analysis been performed appropriately and rigorously? 

Reviewer #1: N/A

Reviewer #2: Yes

Reviewer #3: Yes

4. Have the authors made all data underlying the findings in their manuscript fully available?

Reviewer #1: Yes

Reviewer #2: Yes

Reviewer #3: Yes

5. Is the manuscript presented in an intelligible fashion and written in standard English?

Reviewer #1: Yes

Reviewer #2: Yes

Reviewer #3: Yes

6. Review Comments to the Author

Reviewer #1: The manuscript has been substantially improved. With the exception of a few suggestions, below, I think it is ready to go.

Line 130. Probably should be “liters”.

Line 142. “10-quart” should be given in metric units.

Line 337. Probably should be “few or no coals”

Line 84, 113, 121 (Discussion) Probably should be “coarse”

Reviewer #2: I am reviewing this contribution for the second time. I see that the authors have been mindful and thorough about responding to previous reviews. They have also generously made all data, including sample-OTU tables and scripts available through appropriate depositories.

It is true that some of the reviewer comments are impossible to address without redoing the entire experiment – or repeating for additional blocks. That said, I fully agree with the authors, point of the contribution is not to describe an experiment to infer responses to treatment as much as it is to demonstrate the system and provide a proof of concept. I strongly feel that demonstrating heat penetration and organismal responses in a system that permits replicated fire studies is important. I only have a few editorial comments on few issues that may have slipped into the text during the revision.

Line 25: Typo – “We introduce a thermoschemical gradient model to summarize[s] the way that heat”

Line 28: The goldilocks zone is an orphan concept in the abstract. I agree this will be clarified in the forthcoming text, but might need to be omitted or better explained here.

Line 95: “Temperatures from 480-220” – why not “220-480?”

Line 248: Please provide an estimate of rainfall equivalence in mm.

Text below table 1: Pyronema spp. – “spp.” need not be italics

Line 8 in the second round of numbering: Grammar in “was ranked10th the 4th in abundance ”

Reviewer #3: (No Response)

7. PLOS authors have the option to publish the peer review history of their article (what does this mean?). If published, this will include your full peer review and any attached files.

Reviewer #1: No

Reviewer #2: No

Reviewer #3: No

---

## [Author Response · Author response to Decision Letter 1]

12 Feb 2020

all minor edits have been made.

 this line is now added at line 168 "All soils were collected in Spring 2015, under Special Use Permit #GRO1087 from the USDA Forest Service, Stanislaus National Forest to TDB. "

---

## [Editor Report · Decision Letter 2]

14 Feb 2020

A simple pyrocosm for studying soil microbial response to fire reveals a rapid, massive response by Pyronema species

PONE-D-19-24954R2

Dear Dr. Bruns,

We are pleased to inform you that your manuscript has been judged scientifically suitable for publication and will be formally accepted for publication once it complies with all outstanding technical requirements.

With kind regards,

Garret Suen, Ph.D.

Academic Editor

PLOS ONE
---

## [Editor Report · Acceptance letter]

19 Feb 2020

PONE-D-19-24954R2 

A simple pyrocosm for studying soil microbial response to fire reveals a rapid, massive response by *Pyronema* species 

Dear Dr. Bruns:

I am pleased to inform you that your manuscript has been deemed suitable for publication in PLOS ONE. Congratulations! Your manuscript is now with our production department. 

With kind regards,

on behalf of

Dr. Garret Suen 

Academic Editor

PLOS ONE